# Combating the Instability of Mutual Information-based Losses via Regularization

**Kwanghee Choi**[*1]                    **Siyeong Lee**[*2]

[1]Sogang University
[2]NAVER LABS

## Abstract

Notable progress has been made in numerous fields of machine learning based on neural network-driven mutual information (MI) bounds. However, utilizing the conventional MI-based losses is often challenging due to their practical and mathematical limitations. In this work, we first identify the symptoms behind their instability: (1) the neural network not converging even after the loss seemed to converge, and (2) saturating neural network outputs causing the loss to diverge. We mitigate both issues by adding a novel regularization term to the existing losses. We theoretically and experimentally demonstrate that added regularization stabilizes training. Finally, we present a novel benchmark that evaluates MI-based losses on both the MI estimation power and its capability on the downstream tasks, closely following the pre-existing supervised and contrastive learning settings. We evaluate six different MI-based losses and their regularized counterparts on multiple benchmarks to show that our approach is simple yet effective.

## 1 INTRODUCTION

Identifying a relationship between two variables of interest is one of the key problems in mathematics, statistics, and machine learning [Goodfellow et al., 2014, Ren et al., 2015, He et al., 2016, Vaswani et al., 2017]. One of the fundamental approaches is information theory-based measurement, namely the measure of mutual information (MI). Due to its mathematical soundness and the rise of deep learning, many have designed differentiable MI-based losses for neural networks. Some utilize the MI-based losses to bridge the gap between latent variables and representations in generative adversarial networks [Nowozin et al., 2016, Chen et al.,

2016, Belghazi et al., 2018, van den Oord et al., 2018, Hjelm et al., 2019], where others introduce MI-based methodologies identifying the relationship between input, output, and hidden variables [Tishby and Zaslavsky, 2015, Shwartz-Ziv and Tishby, 2017, Saxe et al., 2018]. Furthermore, recent self-supervised losses use contrastive losses, where its origin can be traced back to MI-based losses [Cheng et al., 2020, Hénaff, 2020, Chuang et al., 2020].

Although many have shown computational tractability and usefulness of MI-based losses, others still struggle with their instability during optimization. Contrastive learning literature with MI-based losses such as Chen et al. [2020], He et al. [2020] use huge batch sizes to reduce the variance of losses. Bardes et al. [2021] adds a regularization term to the neural network embeddings to stabilize the training. McAllester and Stratos [2020] and Song and Ermon [2020] further provide theoretical limitations of variational MI estimators, arguing that the limited batch size induces a MI estimation variance too large to handle. We argue that mitigating the variance of MI-based losses is critical for stabilizing training, where it is well known that more stable optimization of neural networks yields better predictive performance on the downstream tasks [Rothfuss et al., 2019, Bear and Cushman, 2020, Chavdarova et al., 2019, Richter et al., 2020, Zeng et al., 2020, Colombo et al., 2021].

In this paper, we concentrate on identifying the cause behind the instability of MI-based losses and propose a simple yet effective regularization method that can be applied to various MI-based losses. We start by analyzing the behaviors of two MI estimators; the MI Neural Estimator (MINE) loss [Belghazi et al., 2018] and Nguyen-Wainwright-Jordan loss (NWJ) loss [Nguyen et al., 2010]. We identify two distinctive behaviors that induce instability during training, drifting and exploding neural network outputs. Based on these observations, we design two novel dual representations of the KL-divergence called Regularized Donsker-Varadhan representation (ReDV) and Regularized NWJ representation (ReNWJ). We show theoretically and experimentally that adding our regularizer term suppresses two behaviors

---

*These authors contributed equally to this work.

*Accepted for the 38th Conference on Uncertainty in Artificial Intelligence* (UAI 2022).

of drifting and exploding, avoiding instability during training. Finally, we design a novel benchmark that bridges the gap between variational MI estimators and real-world tasks, whereas previous works either do not directly show the MI estimation performance or evaluate only on toy problems. We reformulate both the supervised and the contrastive learning problem [Chen et al., 2020, He et al., 2020, Khosla et al., 2020] as MI estimation problems and show that our regularization yields better performance on both perspectives, downstream task and MI estimation performance.

## 2 BACKGROUND & RELATED WORKS

**Definition of MI** The mutual information between two random variables $X$ and $Y$ is defined as

$$
\begin{aligned}
I(X,Y) &= D_{\text{KL}}(\mathbb{P}_{XY}||\mathbb{P}_X \otimes \mathbb{P}_Y) \\
&= \mathbb{E}_{\mathbb{P}_{XY}}(\log \frac{d\mathbb{P}_{XY}}{d\mathbb{P}_{X\otimes Y}})
\end{aligned} \quad (1)
$$

where $\mathbb{P}_{XY}$ and $\mathbb{P}_X \otimes \mathbb{P}_Y$ are the joint distribution and the product of the marginal distributions, respectively. $D_{\text{KL}}$ is the Kullback-Leibler (KL) divergence. Without loss of generality, we consider $\mathbb{P}_{XY}$ and $\mathbb{P}_X \otimes \mathbb{P}_Y$ as being distributions on a compact domain $\Omega \subset \mathbb{R}^d$.

**MI through dual representation of $D_{\text{KL}}$** We first introduce two dual representations of $D_{\text{KL}}$, as MI is defined using $D_{\text{KL}}$. The most widely known is the Donsker-Varadhan representation $D_{\text{DV}}$ [Donsker and Varadhan, 1975]. For given two distribution $\mathbb{P}$ and $\mathbb{Q}$ on some compact domain $\Omega \subset \mathbb{R}^d$,

$$
D_{\text{DV}}(X,Y) := \sup_{T:\Omega\to\mathbb{R}} \mathbb{E}_{\mathbb{P}}(T) - \log(\mathbb{E}_{\mathbb{Q}}(e^T)), \quad (2)
$$

where both the expectations $\mathbb{E}_{\mathbb{P}}(T)$ and $\mathbb{E}_{\mathbb{Q}}(e^T)$ are finite. If we substitute $\mathbb{P}$ and $\mathbb{Q}$ into $\mathbb{P}_{XY}$ and $\mathbb{P}_X \otimes \mathbb{P}_Y$, $D_{\text{DV}}$ yields the definition of MI. The optimal $T^* = \log \frac{d\mathbb{P}}{d\mathbb{Q}} + C$, where $C \in \mathbb{R}$ can be any constant.

In contrast to $D_{\text{DV}}$, the Nguyen-Wainwright-Jordan representation $D_{\text{NWJ}}$ [Nguyen et al., 2010] is induced by the convex conjugate known as Fenchel's inequality [Hiriart-Urruty and Lemaréchal, 2004]:

$$
D_{\text{NWJ}}(X,Y) := \sup_{T:\Omega\to\mathbb{R}} \mathbb{E}_{\mathbb{P}}(T) - \mathbb{E}_{\mathbb{Q}}(e^{T-1}) \quad (3)
$$

The optimal $T^* = \log \frac{d\mathbb{P}}{d\mathbb{Q}} + 1$ is unique unlike the optimal $T^*$ of $D_{\text{DV}}$ due to its self-normalizing property [Belghazi et al., 2018]. However, $D_{\text{DV}}$ guarantees tighter lower bounds than $D_{\text{NWJ}}$ [Ruderman et al., 2012, Polyanskiy and Wu, 2014]. These two representations provide the theoretical soundness for numerous variational MI bounds.

**Variational MI estimation** With the increasing success of neural networks, several neural network-driven variational bounds of MI are proposed. They are widely employed, such

as contrastive learning [van den Oord et al., 2018, He et al., 2020, Chen et al., 2020] or generative adversarial training [Belghazi et al., 2018, Nowozin et al., 2016]. Variational bounds of MI commonly focus on estimating $T^*$ via a neural network $T_\theta : \Omega \to \mathbb{R}$, called the statistics network [Belghazi et al., 2018], which outputs a single real value given the input sample pairs.

$I_{\text{MINE}}$ [Belghazi et al., 2018] directly maximize $D_{\text{DV}}$ as the objective function by feeding the samples $(x,y)$ of $\mathbb{P}_{XY}$ and $\mathbb{P}_X \otimes \mathbb{P}_Y$ into $T_\theta$:

$$
\begin{aligned}
I_{\text{MINE}}(X,Y) := \\
\mathbb{E}_{\mathbb{P}_{XY}^{(n)}}(T_\theta(x,y)) - \log(\mathbb{E}_{\mathbb{P}_X^{(n)}\otimes\mathbb{P}_Y}^{(n)}(e^{T_\theta(x,y)})), \quad (4)
\end{aligned}
$$

where $\mathbb{P}^{(n)}$ is the empirical distribution associated to $n$ i.i.d. samples for given distribution $\mathbb{P}$. Belghazi et al. [2018] also utilizes moving averages of mini-batches to reduce the MI estimation variance caused by the limited batch size.

$I_{\text{InfoNCE}}$ [van den Oord et al., 2018] is also commonly used due to its stability and decent performance:

$$
I_{\text{InfoNCE}}(X,Y) = \frac{1}{N}\sum_{i=1}^{N} \log \frac{e^{T_\theta(x_i,y_i)}}{\frac{1}{N}\sum_j^N e^{T_\theta(x_i,y_j)}} \quad (5)
$$

where the $N$ samples $(x_i,y_i)_{i=1}^N$ are drawn from $\mathbb{P}_{XY}$, which becomes equivalent to using the Softmax function with the negative log loss. $I_{\text{InfoNCE}}$ is also equivalent to $I_{\text{MINE}}$ up to a constant, but upper bounded by $\log N$, hence not able to estimate large MI values [van den Oord et al., 2018].

Poole et al. [2019] introduced $I_{\text{TUBA}}$, a unified lower bound, by expanding $D_{\text{NWJ}}$ [Barber and Agakov, 2003, Nguyen et al., 2010].

$$
\begin{aligned}
I_{\text{NWJ}}(X,Y) := \\
\mathbb{E}_{\mathbb{P}_{XY}^{(n)}}(T_\theta(x,y)) - \mathbb{E}_{\mathbb{P}_X^{(n)}\otimes\mathbb{P}_Y^{(n)}}(e^{T_\theta(x,y)-1}), \quad (6)
\end{aligned}
$$

$$
\begin{aligned}
I_{\text{TUBA}}(X,Y) := \mathbb{E}_{\mathbb{P}_{XY}^{(n)}}(T_\theta(x,y)) \\
- \mathbb{E}_{\mathbb{P}_Y^{(n)}}\left(\mathbb{E}_{\mathbb{P}_X^{(n)}}(e^{T_\theta(x,y)})/a(y) + \log(a(y)) - 1\right), \quad (7)
\end{aligned}
$$

where $a(y)$ is the variational parameter. However, unlike $I_{\text{MINE}}$ or $I_{\text{InfoNCE}}$, directly using the exponential term often causes numerical instability. Even if $T_\theta$ outputs a moderately sized value, $e^{T_\theta}$ can easily exceed the floating-point range.

To avoid this problem, Poole et al. [2019] introduce $D_{\text{NWJ}}$-based lower bound $I_{\text{JS}}$ by using a softplus-activated neural network as $T_\theta$,

$$
\begin{aligned}
I_{\text{JS}}(X,Y) := 1 + \mathbb{E}_{\mathbb{P}_{XY}^{(n)}}(T_\theta(x,y)) \\
- \mathbb{E}_{\mathbb{P}_Y^{(n)}\otimes\mathbb{P}_X^{(n)}}((e^{T_\theta(x,y)})). \quad (8)
\end{aligned}
$$

**Variance problem of MI estimators** Despite the variety of bounds proposed, many still suffer from the bias-variance trade-off [Poole et al., 2019]. McAllester and Stratos [2020] and Song and Ermon [2020] prove that the $I_{\text{MINE}}$ estimator must have a batch size proportional to the exponential of true MI to control the variance of the estimation.

Many bounds try to mitigate this problem by reducing the variance of low-biased estimators, such as by interpolating with a low variance bound [Poole et al., 2019] or dropping the formal theoretical guarantees [McAllester and Stratos, 2020]. Song and Ermon [2020] proposed $I_{\text{SMILE}}$ to clip the range of $T_\theta$ trained with $I_{\text{MINE}}$, sacrificing the estimation quality to reduce the variance.

$$
\begin{aligned}
I_{\text{SMILE}}(X, Y) := \ & \mathbb{E}_{\mathbb{P}_{XY}^{(n)}} (T_\theta(x, y)) \\
& - \log(\mathbb{E}_{\mathbb{P}_X^{(n)} \otimes \hat{\mathbb{P}}_Y^{(n)}} (\text{clip}(e^{T_\theta(x,y)}, e^{-\tau}, e^{\tau}))), \quad (9)
\end{aligned}
$$

where $\text{clip}(v, l, u) = \max(\min(v, u), l)$ for $v, u, l \in \mathbb{R}$.

**Practical usages of MI** MI-based losses are often applied in generative modeling, such as for better mode coverage [Belghazi et al., 2018] or learning disentangled representations without supervision [Chen et al., 2016, Ojha et al., 2020, Li et al., 2021b, Jeon et al., 2021]. Representation learning employs MI-based losses [Tian et al., 2020b, Hjelm et al., 2019, Tschannen et al., 2020, Cheng et al., 2020, Wu et al., 2020, Wen et al., 2020, Boudiaf et al., 2020, Tian et al., 2020a, Li et al., 2021a] to yield feature extractors that reflect its downstream tasks well. We emphasize that these approaches can be further utilized to measure the performance of MI estimators.

**Comparing between MI estimators** Toy datasets such as correlated multivariate Gaussian distributions has been widely accepted for the evaluation of MI estimation [Belghazi et al., 2018, Poole et al., 2019, Song and Ermon, 2020, Cheng et al., 2020, Lin et al., 2019]. However, we emphasize that using synthetic data as a definitive benchmark will end up in a disparity with real-world tasks. There have been some approaches that compared different MI estimators on generative modeling [Belghazi et al., 2018, Hjelm et al., 2019] or representational learning [Tian et al., 2020b]. However, finding the ideal MI for each downstream task is not trivial, making it impossible to directly assess the MI estimation quality. Moreover, Tschannen et al. [2020] and Tian et al. [2020b] showed the gap between MI estimation quality and downstream performance on specific tasks. Hence, it is crucial to evaluate both perspectives. The closest work to our benchmark is the consistency test of Song and Ermon [2020] using CIFAR-10 [Krizhevsky, 2009] and MNIST [LeCun et al., 1998]. However, the test only offered to assess the ratio of two separate MI estimations, making it difficult to separately measure the quality of each estimation.

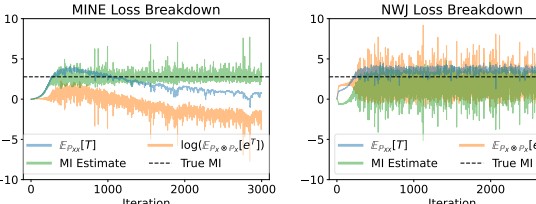

Figure 1: Training $T_\theta$ using $I_{\text{MINE}}$ and $I_{\text{NWJ}}$ with batch size 100 for 3000 iterations. We breakdown the MI loss into two components. We split $I_{\text{MINE}}$ into first term $\mathbb{E}_{\mathbb{P}_{XX}}(T)$ and second term $\log \mathbb{E}_{\mathbb{P}_X \otimes \mathbb{P}_X}(e^T)$. Similarly, we split $I_{\text{NWJ}}$ into first term $\mathbb{E}_{\mathbb{P}_{XX}}(T)$ and second term $\mathbb{E}_{\mathbb{P}_X \otimes \mathbb{P}_X}(e^{T-1})$.

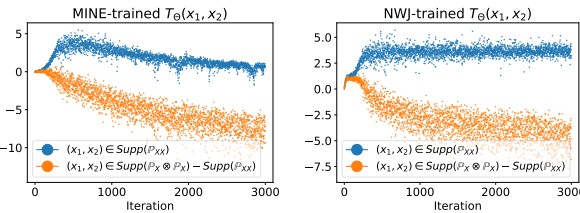

Figure 2: Training $T_\theta$ using $I_{\text{MINE}}$ and $I_{\text{NWJ}}$ with batch size 100 for 3000 iterations. We observe the statistics network outputs $T_\theta(x_1, x_2)$, where we split the outputs into two: $(x_1, x_2) \in Supp(\mathbb{P}_{XX})$ and $(x_1', x_2') \in Supp(\mathbb{P}_X \otimes \mathbb{P}_X) \setminus Supp(\mathbb{P}_{XX})$.

## 3 INSTABILITY OF MI BOUNDS

To demonstrate and analyze the instability of variational MI bounds, we design a synthetic problem with the One-hot dataset. We then solve the task via $I_{\text{MINE}}$ and $I_{\text{NWJ}}$, which are the losses derived from the two most commonly used representations of KL-divergence, $D_{\text{DV}}$ and $D_{\text{NWJ}}$, respectively. Both losses consist of two terms, each derived from the statistics of joint distribution $\mathbb{E}_{\mathbb{P}_{XY}}$ and the product of marginal distributions $\mathbb{E}_{\mathbb{P}_X \otimes \mathbb{P}_Y}$. Hence, to observe the behavior of each loss during training, we plot the two terms separately. Also, to observe how each distribution differ by the statistics network outputs $T_\theta(x, y)$, we plot each output from $(x, y) \sim Supp(\mathbb{P}_{XY})$ and $(x, y) \sim Supp(\mathbb{P}_X \otimes \mathbb{P}_Y) \setminus Supp(\mathbb{P}_{XY})$, where we denote the support of $\mathbb{P}$ as $Supp(\mathbb{P})$. Support is the set of values that the random variable can take [Taboga, 2021].

**One-hot Dataset** We design a one-hot discrete problem with uniform distribution $X \sim U(1, N)$ to estimating $I(X, X) = \log N$ for a given integer $N$. This task is intentionally created to easily discern samples $(x, x) \sim \mathbb{P}_{XX}$ from $(x, x) \sim \mathbb{P}_X \otimes \mathbb{P}_X$, so that we can directly observe its network outputs $T_\theta(x, x)$.

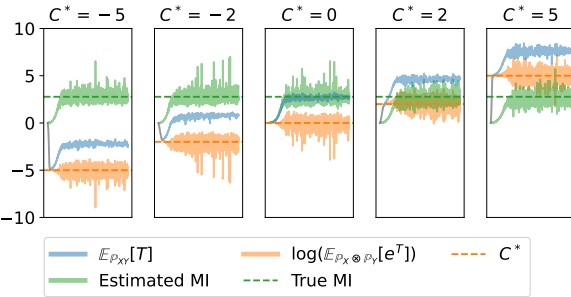

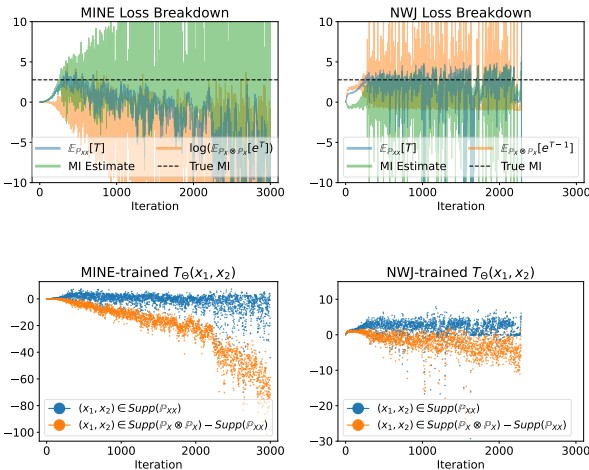

Figure 3: Training $T_\theta$ using $I_{\text{MINE}}$ and $I_{\text{NWJ}}$ with a reduced batch size of 32 for 3000 iterations. MI estimate diverges for both cases. Also, $I_{\text{NWJ}}$ incurs exploding $T_\theta$ outputs, hence the empty plot after 23k iterations.

Figure 4: Training $T_\theta$ with batch size 100 for 1500 iterations using $I_{\text{ReMINE}}$ with different $C^*$ (orange dotted line).

**Seemingly Stable Case** We first observe the behaviors of the statistics network $T_\theta$ when the losses are seemingly stable, producing a successful MI estimate. Fig. 1 shows the MI estimates and the two terms that construct each MI estimate per batch. We observe that the first and the second term estimates of $I_{\text{MINE}}$, unlike $I_{\text{NWJ}}$, drifting in parallel even after the MI estimate converge. This is due to the free constant term $C$ in the optimal $T^*$ of $D_{\text{DV}}$, where the self-normalizing $D_{\text{NWJ}}$ avoids this problem. This drifting phenomenon implies that $T_\theta$ is not stable even after the loss seems to be converged, as shown in Fig. 2. Also, the plot demonstrates how $T_\theta$ is trained; it isolates the samples $(x, y) \sim \mathbb{P}_{XY}$ from the samples $(x, y) \sim \mathbb{P}_X \otimes \mathbb{P}_Y$.

**Unstable Case** We also demonstrate the behaviors of $T_\theta$ when the losses get unstable in Fig. 3. We reduce the batch size to make the optimization unstable, where this behavior is often reported in multiple works [van den Oord et al., 2018, He et al., 2020, Chen et al., 2020]. However, even though the losses seem unstable, $T_\theta$ successfully discerns the samples before the outputs explode. We believe that this is because of how $T_\theta$ is optimized during training. The statistics network outputs $T_\theta(x_1, x_2)$ of $(x_1, x_2) \in Supp(\mathbb{P}_{XX})$ gets increased by the first term but occasionally decreased by the second term. However, $T_\theta(x_1', x_2')$ of $(x_1', x_2') \in Supp(\mathbb{P}_X \otimes \mathbb{P}_X) \setminus Supp(\mathbb{P}_{XX})$ gets decreased whatsoever, as $(x_1', x_2')$ is used only for the second term. This makes the second term more unstable and motivates us to regularize it for better numerical stability during optimization.

To summarize, we suspect the instability of variational bounds comes from two reasons. Firstly, the statistics net-

work did not converge even after the loss seemingly converged. We argue that this is due to the unnormalized constant term in the optimal $T^*$ of $D_{\text{DV}}$, where $D_{\text{NWJ}}$ successfully avoids via self-normalization. Secondly, the loss gets unstable as $T_\theta(x_1', x_2')$ endlessly decrease due to the second term. This observation is also consistent with the theoretical findings of Song and Ermon [2020], McAllester and Stratos [2020], where they show that large variance of the second term leads to failed MI estimation. We claim that the outputs have to be regularized in some form to avoid the instability.

## 4 STABILIZING THE MI BOUNDS

In this section, we introduce two novel regularized representations and its corresponding losses to tackle the instability during optimization. We show both theoretically and experimentally that adding regularization mitigates the unstable behavior of the statistics network $T_\theta$. We also describe a simple windowing method that can sidestep the batch size limitation problem of the MI estimation problem. We defer all the proofs to the Appendix.

**Regularized representations** We stabilize the two existing representations $D_{\text{DV}}$ and $D_{\text{NWJ}}$ by regularizing the second term. We introduce two novel representations: Regularized DV ($D_{\text{ReDV}}$) and Regularized NWJ ($D_{\text{ReNWJ}}$),

$$
\begin{aligned}
D_{\text{ReDV}}(X, Y) := \sup_{T:\Omega \to \mathbb{R}} & \mathbb{E}_{\mathbb{P}}(T) - \log(\mathbb{E}_{\mathbb{Q}}(e^T)) \\
& - d(\log(\mathbb{E}_{\mathbb{Q}}(e^T)), C^*),
\end{aligned}
\tag{10}
$$

$$
\begin{aligned}
D_{\text{ReNWJ}}(X, Y) := \sup_{T:\Omega \to \mathbb{R}} & \mathbb{E}_{\mathbb{P}}(T) - \mathbb{E}_{\mathbb{Q}}(e^{T-1})) \\
& - d(\mathbb{E}_{\mathbb{Q}}(e^{T-1}), 1),
\end{aligned}
\tag{11}
$$

where $C^* \in \mathbb{R}$ is any constant and $d(*, *)$ is a distance function on $\mathbb{R}$.

**Theorem 1.** $D_{\text{ReDV}}$ and $D_{\text{ReNWJ}}$ is a dual representation for $D_{KL}$ such that

$$
D_{KL}(\mathbb{P}||\mathbb{Q}) = D_{\text{ReDV}}(X, Y), \tag{12}
$$

$$
D_{KL}(\mathbb{P}||\mathbb{Q}) = D_{\text{ReNWJ}}(X, Y). \tag{13}
$$

We emphasize that both representations are not MI-specific but dual representations of $D_{\mathrm{KL}}$, which can be easily extended to numerous variational MI bounds based on $D_{\mathrm{DV}}$ and $D_{\mathrm{NWJ}}$. Especially, the newly added regularizer grants $D_{\mathrm{ReDV}}$ the normalizing property, effectively solving the drifting problem of $D_{\mathrm{DV}}$.

**Regularizing $I_{\mathbf{MINE}}$ and $I_{\mathbf{NWJ}}$** Based on $D_{\mathrm{ReDV}}$ and $D_{\mathrm{ReNWJ}}$, we propose a novel neural network-driven variational MI bound $I_{\mathrm{ReMINE}}$ and $I_{\mathrm{ReNWJ}}$ by choosing the Euclidean distance $d(x, y) = (x - y)^2$ and the log-Euclidean distance $d(x, y) = (\log x - \log y)^2$, respectively.

$$
\begin{aligned}
I_{\mathrm{ReMINE}}(X,Y) := {}& \mathbb{E}_{\mathbb{P}_{XY}^{(n)}}\left(T_\theta(x, y)\right) \\
& - \log\left(\mathbb{E}_{\mathbb{P}_X^{(n)} \otimes \mathbb{P}_Y^{(n)}}\left(e^{T_\theta(x,y)}\right)\right) \\
& - \lambda\left(\log\left(\mathbb{E}_{\mathbb{P}_X^{(n)} \otimes \mathbb{P}_Y^{(n)}}\left(e^{T_\theta(x,y)}\right)\right) - C^*\right)^2,
\end{aligned}
\tag{14}
$$

$$
\begin{aligned}
I_{\mathrm{ReNWJ}}(X,Y) := {}& \mathbb{E}_{\mathbb{P}_{XY}^{(n)}}\left(T_\theta(x, y)\right) \\
& - \mathbb{E}_{\mathbb{P}_X^{(n)} \otimes \mathbb{P}_Y^{(n)}}\left(e^{T_\theta(x,y)-1}\right) \\
& - \lambda\left(\log\left(\mathbb{E}_{\mathbb{P}_X^{(n)} \otimes \mathbb{P}_Y^{(n)}}\left(e^{T_\theta(x,y)-1}\right)\right)\right)^2,
\end{aligned}
\tag{15}
$$

where $C^* \in \mathbb{R}$ is any constant and $\lambda$ is a hyperparameter that controls the degree of regularization. We can also easily regularize other losses such as $I_{\mathrm{InfoNCE}}$, $I_{\mathrm{SMILE}}$, $I_{\mathrm{TUBA}}$, and $I_{\mathrm{JS}}$ in a plug-and-play manner. See Table 1 for more details on its regularized counterparts.

**Solving the drifting problem** Due to the self-regularizing nature of $D_{\mathrm{NWJ}}$, we must fix $C^* = 1$ for $I_{\mathrm{ReNWJ}}$. We also set $C^* = 0$ for $I_{\mathrm{ReMINE}}$ on future experiments, but to demonstrate the ability of the regularizer term to stop the drifting, we experiment with various $C^*$ in Fig. 4. Comparing to $I_{\mathrm{MINE}}$ in Fig. 1, we can observe that $I_{\mathrm{ReMINE}}$ successfully solves the drifting problem by regularizing the second term to have a single value.

**Solving the explosion problem** We previously observed the instability of $I_{\mathrm{MINE}}$ and $I_{\mathrm{NWJ}}$ when using a small batch in Fig. 3. We apply the same setting to $I_{\mathrm{ReMINE}}$ and $I_{\mathrm{ReNWJ}}$ to observe if the regularizer mitigates the instability problem. Both regularized losses successfully avoid the explosion problem and limit the statistics network outputs $T_\theta(x_1, x_2)$ within a certain boundary. As discussed in Section 3, the second term was the culprit of the variance in MI estimation. The newly added term directly regularizes it to stabilize training, giving the statistics network $T_\theta$ additional hints for the second term to converge to a specific value $C^*$ successfully. Furthermore, we empirically found that our regularization works well with $I_{\mathrm{SMILE}}$'s strategy of clipping $T_\theta$. Gradient zeros out for the original $I_{\mathrm{SMILE}}$ if $T_\theta(x, y)$ exceeds a certain threshold. This behavior makes $T_\theta$ act as if it were frozen, failing to further optimize during training. However, with the regularizer term, we can clip $T_\theta(x, y)$

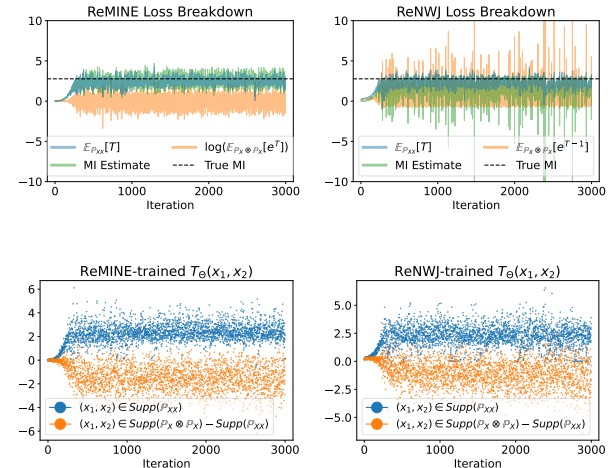

Figure 5: Training $T_\theta$ using the regularized counterparts, $I_{\mathrm{ReMINE}}$ and $I_{\mathrm{ReNWJ}}$, with the same small batch settings from Fig. 3. Regularization effectively mitigates both instability symptoms, shifting and exploding.

only on first and second term, i.e., on the original loss. Now, clipping filters out the noisy gradients while the gradients calculated from the regularizer avoid freezing $T_\theta$ entirely.

**Mathematical properties of $I_{\mathbf{ReMINE}}$ and $I_{\mathbf{ReNWJ}}$** Following Belghazi et al. [2018], we show the soundness of $I_{\mathrm{ReMINE}}$ and $I_{\mathrm{ReNWJ}}$ in two perspectives, strong consistency and sample complexity. These properties relate to whether the trained $T_\theta$ can be sufficiently similar to the optimal $T^*$.

**Theorem 2.** *$I_{ReMINE}$ and $I_{ReNWJ}$ are strongly consistent.*

For the two losses, we also provide the mathematical bound on the number of samples required for the empirical MI estimation at a given accuracy and with high confidence. Similar to Belghazi et al. [2018], let $T_\theta$ satisfy $L$-Lipschitz with respect to the parameter $\theta$ such that $|\theta| < K$ and $d$ is dimension of the parameter space of $T_\theta$.

**Theorem 3.** *Assume that $T_\theta$ is bounded above by $M$. Let $k$ be the number of sample means. Given any $\epsilon, \delta$ of the desired accuracy and confidence parameters, we have*

$$
\mathcal{P}(|I_{ReMINE}(X;Y) - I(X,Y)| \leq \epsilon) \geq 1 - \delta, \tag{16}
$$

*whenever the number $n$ of samples satisfies*

$$
n \geq \frac{d\log(24KL\sqrt{d}/\epsilon) + 2dM + \log(2/\delta)}{\epsilon^2 k/(2M^2)}. \tag{17}
$$

**Theorem 4.** *Assume that $1 \leq |T_\theta| < M$ and $d(x, 1) \leq |x - 1|$. Let $k$ be the number of sample means. Given any $\epsilon, \delta$ of the desired accuracy and confidence parameters, we have*

$$
\mathcal{P}(|I_{ReNWJ}(X;Y) - I(X,Y)| \leq \epsilon) \geq 1 - \delta, \tag{18}
$$

*whenever the number $n$ of samples satisfies*

$$n \geq \frac{d\log(24KL\sqrt{d}/\epsilon) + 2dM + \log(2/\delta)}{\epsilon^2 k/(2M^2)}. \quad (19)$$

**Drifting may lead to noisy MI estimate**  We prove that the variance of the second term on the empirical distributions is affected by the constant term $C^*$.

**Theorem 5.** *Let $Q^{(n)}$ be the empirical distributions of $n$ i.i.d. samples from $\mathbb{Q}$. For the optimal $T_1 = \log\frac{dp}{dq} + C_1$ and $T_2 = \log\frac{dp}{dq} + C_2$ where $C_1 \geq C_2$,*

$$Var_{\mathbb{Q}}(\mathbb{E}_{\mathbb{Q}^{(n)}}(e^{T_1})) \geq Var_{\mathbb{Q}}(\mathbb{E}_{\mathbb{Q}^{(n)}}(e^{T_2})) \quad (20)$$

This implies that unregulated $C^*$ may lead to worse MI estimation quality, as the source of the estimate variance are mainly due to the second term.

**Increasing the effective sample size for MI estimation**
The drifting problem caused by the unnormalized constant term $C^*$ raises more issues when estimating MI. Poole et al. [2019] use a simple macro-averaging technique, i.e., averaging the estimated MI from each batch. We can also consider a slight modification to the technique, where we call it the micro-averaging technique, by saving all the statistics network outputs $T_\theta(x, y)$ for each batch and producing a single estimate based on all the outputs. However, we proved that both averaging techniques yield wrong final estimates for biased estimators like $I_{\text{MINE}}$ [Belghazi et al., 2018], $I_{\text{SMILE}}$ [Song and Ermon, 2020], $I_{\text{CLUB}}$ [Cheng et al., 2020], and $I_{\text{InfoNCE}}$ [van den Oord et al., 2018].

**Theorem 6.** *(Estimation bias caused by drifting)  Both macro- and micro-averaging strategies produce a biased MI estimate when the drifting problem occurs.*

To the contrary, self-normalizing or regularized MI estimators have the upper hand in this perspective. By utilizing all the samples from multiple batches, they can effectively sidestep the batch size limitation problem [McAllester and Stratos, 2020, Song and Ermon, 2020].

## 5 EXPERIMENTS

### 5.1 MI ESTIMATION VS. DOWNSTREAM TASK PERFORMANCE

**Benchmark Design**  To measure the performance of the MI estimators, one must design the target task to have the ground truth MI. This constraint led previous works to evaluate the estimators only on artificial toy problems [Belghazi et al., 2018, Poole et al., 2019], where its connection to actual problems is fairly limited. We design the two types of MI estimation tasks with de facto image datasets to improve the existing benchmarks to reflect on the real-world tasks. We defer all the proofs to the Appendix.

**Theorem 7.** *(Supervised learning) Given a dataset $D = (X, Y)$ where $X$ is an sample, $Y$ is the label for $X$, and $H(Y)$ is the entropy of the label set, $I(X, Y) = H(Y)$.*

Similarly, the true MI between images from the same class is also tractable based on the same assumption.

**Theorem 8.** *(Contrastive learning) Consider the dataset $D = (X, Y)$. Let $X_1$ be a sample drawn from the dataset and $X_2$ be another sample drawn from the subset with the same label $Y$ to $X_1$. Then, $I(X_1, X_2) = I(X_1, Y) = I(X_2, Y) = H(Y)$.*

Note that we assume statistical dependence between the image $X$ and label $Y$ from the point of view of information bottleneck [Tishby and Zaslavsky, 2015]. We derive the theorems above based on the assumption, where $Y$ implicitly determines $X$.

Based on the above theorems, we use the two MI estimation problems as benchmarks that evaluate the performance of estimators. We intentionally design Theorem 7 and Theorem 8 to mimic the existing tasks closely, namely, supervised and contrastive learning. For Theorem 7, we can set the statistics network $T_\theta(X, Y) = f_\theta(X) \cdot o(Y)$ where $f_\theta(X)$ is the logits obtained from feeding the image $X$ to the classification neural networks and $o(Y)$ is the one-hot representation of the label $Y$. If we use the InfoNCE estimator, this formulation becomes identical to solving the classification problem using negative log loss with the Softmax function, hence the name being the supervised learning benchmark (SLB). Similarly, for Theorem 8, we can set $T_\theta(X_1, X_2) = f_\theta(X_1) \cdot f_\theta(X_2)$ and use the InfoNCE estimator to yield a commonly used contrastive loss [van den Oord et al., 2018, Chen et al., 2020].

Due to the strict assumption of statistical dependence, the theorems above cannot be used on standard datasets like ImageNet dataset [Deng et al., 2009], as its samples often violate the single-label assumption. However, we can still empirically compare the MI estimators by the relative size of their final MI estimation. We conduct a demo experiment on ImageNet in the Appendix.

**Evaluation**  To verify the performance of MI estimators, we perform our benchmark tasks on the CIFAR10 and CIFAR100 dataset [Krizhevsky, 2009]. As both CIFAR10 and CIFAR100 have a uniform label distribution, ideal MI is $\log 10$ and $\log 100$, respectively. In addition, to check whether this MI estimate task is actually helpful for downstream tasks, we evaluate each estimator on both dimensions: MI estimation and test set accuracy. Similar to the existing settings in the contrastive learning literature [Chen et al., 2020, He et al., 2020], we design the test accuracy of CLB by defining the label estimate $\hat{y}$ of each test set sample $x_{\text{Test}}$ to be the label of $x = \text{argmax}_{x \in X_{\text{Train}}} f(x) \cdot f(x_{\text{Test}})$ of the train dataset $X_{\text{Train}}$. Similarly for SLB, we chose

| Loss | Loss settings | Regularizer settings |
|---|---|---|
| $I_{\text{MINE}}$ | No gradient moving average | Euclidean distance |
| $I_{\text{SMILE}}$ | Clipping $(\tau = 10)$ | Euclidean distance |
| $I_{\text{InfoNCE}}$ | - | Euclidean distance |
| $I_{\text{NWJ}}$ | - | Log-Euclidean distance, Clipping $(\tau = 10)$ |
| $I_{\text{TUBA}}$ | $a(y) = 1$ | Log-Euclidean distance, Clipping $(\tau = 10)$ |
| $I_{\text{JS}}$ | Estimate with $I_{\text{NWJ}}$ | Euclidean distance |

Table 1: List of MI estimators with its hyperparameters

$\hat{y} = \text{argmax}_y f(x_{\text{Test}}) \cdot o(y)$ where $o(y)$ is the one-hot encoding of $y$. We ran the same experiment 5 times with different seeds to yield a 95% confidence interval.

## 5.2 COMPARISON WITH OUR BENCHMARK

To demonstrate the effectiveness of our novel regularization term, we regularize the two representations, $D_{\text{DV}}$ and $D_{\text{NWJ}}$. We test three realizations for each representation, $I_{\text{MINE}}$ [Belghazi et al., 2018], $I_{\text{SMILE}}$ [Song and Ermon, 2020], and $I_{\text{InfoNCE}}$ [van den Oord et al., 2018] for $D_{\text{DV}}$, $I_{\text{NWJ}}$ [Nguyen et al., 2010], $I_{\text{TUBA}}$ [Poole et al., 2019], and $I_{\text{JS}}$ [Hjelm et al., 2019] for $D_{\text{NWJ}}$. We compare the original losses with its regularized counterparts, a total of $6 \times 2 = 12$. We do not apply averaging scheme on any of the losses, and choose the regularization weight $\lambda \in \{0.1, 0.01, 0.001\}$ that shows the best MI estimation results. See Table 1 for more details.

We observe in Table 2 that additional regularization generally induces better performance on both the MI estimation task and the downstream task (test accuracy). Hence, adding the regularizer to a pre-existing supervised or contrastive learning loss seems to be a viable option to increase the performance further. Even when the performance of the regularized loss slightly degrades, its negative impact is minimal. This implies that even for the case where the regularizer is not greatly helpful, it does not greatly hinder optimization. Especially, it is intriguing that many losses, $I_{\text{MINE}}$, $I_{\text{ReMINE}}$, and $I_{\text{ReTUBA}}$, are better than $I_{\text{InfoNCE}}$ in SLB, which is used as the de facto standard in classification. Also, $I_{\text{SMILE}}$, $I_{\text{NWJ}}$, and $I_{\text{TUBA}}$ fail to converge in CLB, where simply adding a regularization term solves the issue altogether, yielding a competitive or even better performance than all the other losses. Given the fact that numerous contrastive learning literature suffers from instability [Caron et al., 2021, Bardes et al., 2021, Chen et al., 2020, He et al., 2020, Bardes et al., 2021], we emphasize that adding our regularization term can be a simple yet effective method to stabilize training.

Additionally, to observe the impact of regularization strength $\lambda$, we plot the benchmark performance for each $\lambda$ in Fig. 6. We compare the losses on CLB as experimental results suggest that CLB is a more difficult task than SLB, showing significant performance differences between various losses. On CIFAR-10, $\lambda$ acts as a trade-off parameter

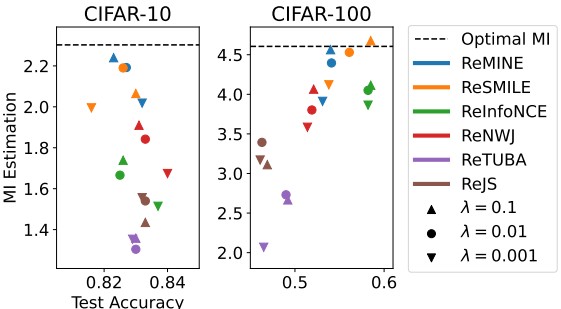

Figure 6: Ablation study on different $\lambda$s with CLB CIFAR-10 and CIFAR-100.

between test accuracy and MI estimation quality. Performance trade-off has also been reported in other literature, where better MI estimation does not necessarily deliver better downstream performance [Tschannen et al., 2020, Tian et al., 2020b]. However, compared to CIFAR-100, test accuracy differences are minimal, where MI differences are apparent. $I_{\text{ReMINE}}$ and $I_{\text{ReSMILE}}$ show excellent MI estimation quality in CIFAR-10 compared to other losses. In contrast, test accuracy and MI estimation quality align well in the CIFAR-100 case. $I_{\text{ReSMILE}}$ shows good overall performance, albeit its sensitivity towards regularization strength. $I_{\text{ReInfoNCE}}$, on the other hand, shows stable performance in the downstream task, sacrificing the MI estimation quality. This result is further supported by the prominence of $I_{\text{InfoNCE}}$ in the contrastive learning domain. It is yet unclear where the difference between CIFAR-10 and CIFAR-100 comes from, whether it is due to the difference in the level of difficulty of the dataset or the batch size used throughout the training. We leave further analysis as future work.

## 5.3 COMPARISON WITH THE STANDARD TOY PROBLEM

We provide the quality of MI-based losses on the 20D Correlated Gaussian task [Belghazi et al., 2018, Poole et al., 2019] where the true MI is increased 5 times during optimization in Fig. 7. This experiment demonstrates how stable the MI-based losses estimate MI in a dynamically changing environment. We apply the same settings from

| Task | | Loss | MI Estimation | | Test Accuracy | |
|---|---|---|---|---|---|---|
| | | | Original | Regularized | Original | Regularized |
| Supervised Learning Benchmark | CIFAR-10 | MINE | **2.300 ± 0.003** | 2.298 ± 0.005 | 0.850 ± 0.009 | **0.856 ± 0.004** |
| | | SMILE | 2.297 ± 0.009 | **2.300 ± 0.003** | **0.854 ± 0.008** | 0.853 ± 0.009 |
| | | InfoNCE | 2.301 ± 0.002 | **2.302 ± 0.001** | 0.845 ± 0.006 | 0.845 ± 0.005 |
| | | NWJ | **2.297 ± 0.009** | 2.294 ± 0.013 | 0.859 ± 0.003 | **0.862 ± 0.004** |
| | | TUBA | 2.297 ± 0.008 | **2.300 ± 0.003** | **0.862 ± 0.008** | 0.859 ± 0.003 |
| | | JS | 1.944 ± 0.039 | **2.000 ± 0.049** | 0.838 ± 0.012 | **0.842 ± 0.004** |
| | CIFAR-100 | MINE | 4.597 ± 0.011 | **4.603 ± 0.001** | 0.610 ± 0.007 | 0.610 ± 0.006 |
| | | SMILE | 4.595 ± 0.015 | **4.602 ± 0.002** | 0.601 ± 0.015 | **0.606 ± 0.007** |
| | | InfoNCE | 4.594 ± 0.017 | **4.599 ± 0.005** | 0.589 ± 0.010 | **0.593 ± 0.005** |
| | | NWJ | 4.572 ± 0.055 | **4.586 ± 0.034** | 0.558 ± 0.042 | **0.599 ± 0.009** |
| | | TUBA | 4.495 ± 0.207 | **4.603 ± 0.002** | 0.543 ± 0.055 | **0.611 ± 0.007** |
| | | JS | 4.088 ± 0.430 | **4.240 ± 0.116** | 0.591 ± 0.026 | **0.598 ± 0.010** |
| Contrastive Learning Benchmark | CIFAR-10 | MINE | 2.233 ± 0.674 | **2.240 ± 0.657** | 0.812 ± 0.026 | **0.823 ± 0.012** |
| | | SMILE | 0.000 ± 0.000 | **2.065 ± 0.842** | 0.100 ± 0.001 | **0.830 ± 0.008** |
| | | InfoNCE | 1.705 ± 0.462 | **1.739 ± 0.431** | **0.830 ± 0.008** | 0.826 ± 0.006 |
| | | NWJ | 0.000 ± 0.000 | **1.910 ± 0.662** | 0.100 ± 0.000 | **0.831 ± 0.005** |
| | | TUBA | 0.000 ± 0.000 | **1.358 ± 0.590** | 0.100 ± 0.000 | **0.830 ± 0.009** |
| | | JS | 1.552 ± 0.485 | **1.556 ± 0.546** | **0.837 ± 0.003** | 0.832 ± 0.009 |
| | CIFAR-100 | MINE | **4.634 ± 0.186** | 4.563 ± 0.162 | 0.522 ± 0.026 | **0.540 ± 0.020** |
| | | SMILE | 0.000 ± 0.000 | **4.677 ± 0.162** | 0.012 ± 0.003 | **0.585 ± 0.007** |
| | | InfoNCE | 4.112 ± 0.147 | **4.115 ± 0.145** | 0.576 ± 0.019 | **0.585 ± 0.014** |
| | | NWJ | 0.000 ± 0.000 | **4.065 ± 0.255** | 0.010 ± 0.000 | **0.521 ± 0.025** |
| | | TUBA | 0.000 ± 0.000 | **2.731 ± 0.786** | 0.010 ± 0.000 | **0.490 ± 0.023** |
| | | JS | 3.253 ± 0.368 | **3.393 ± 0.124** | 0.451 ± 0.020 | **0.463 ± 0.031** |

Table 2: Our supervised and contrastive learning benchmark results. We provide the 95% confidence interval of 5 runs for both MI estimation and test accuracy, where we clip the negative MI estimations to 0. We compare the performance of original and regularized loss. **Bold text** and blue text indicates the better performance with overlapping and non-overlapping confidence interval, respectively.

Table 1, where we fix the regularization strength $\lambda = 1.0$ for all the losses. With the exception of $I_{\text{InfoNCE}}$, regularized losses show clear superiority over the original losses. Regularization facilitates $I_{\text{MINE}}$ and $I_{\text{SMILE}}$ to avoid the instability which is mentioned in Section 3. Also, regularization greatly enhances the MI estimation quality of $I_{\text{JS}}$ and lessens the variance of both $I_{\text{NWJ}}$ and $I_{\text{TUBA}}$.

# 6 CONCLUSION

In this paper, we identify the two symptoms behind the instability: The statistics network was not converging even after the loss seemed to converge, and its outputs from the product of marginal distribution explode during training. We propose a novel regularization term to mitigate the instability during training by adding to various existing MI-based losses. We theoretically and experimentally demonstrate that the added regularizer directly alleviates the two instability symptoms. Finally, we present a benchmark that evaluates both the MI estimation power and its capability on the downstream tasks by imitating the supervised or contrastive learning settings. We compare six different losses and their regularized counterparts on various benchmarks to show the method's effectiveness and broad applicability.

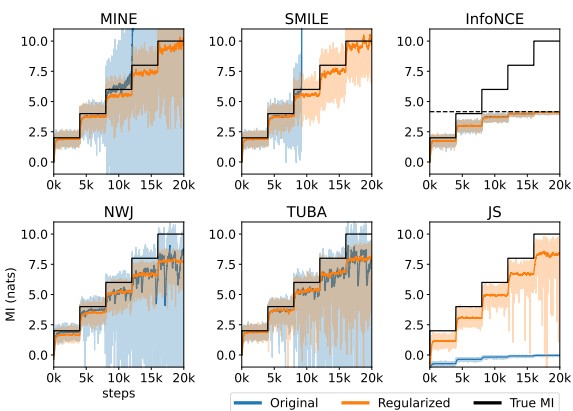

Figure 7: Estimation performance on 20-D Gaussian. The estimated MI (light) and the smoothed estimation with exponential moving average (dark) are plotted for each methods with its regularized counterparts. Black line represents the true MI. Dotted line shows the bound of $I_{\text{InfoNCE}}$ due to the limited batch size of 64.

# LIMITATIONS AND FUTURE WORKS

We suspect that the instability of MI estimators can also be related to the collapse problem [Bardes et al., 2021, Caron et al., 2021]. Further loss-based approaches to combat this problem by regularizing the network outputs may be helpful. We expect that extending our methods to various contrastive learning losses may yield fruitful results for self-supervised learning, notably for other domains such as text or audio. Also, our mathematical analysis is mainly focused on the drifting problem of $I_{\text{MINE}}$, not the explosion problem of $I_{\text{NWJ}}$. For $I_{\text{NWJ}}$, we suspect that the absence of the log function wrapping the exponential values makes the second term much more susceptible to output explosion due to its numerical instability. The added regularizer gives additional hints for the second term to converge to a specific value. However, we did not expand the discussion further in this paper.

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
