# OpenReview forum: "Combating the Instability of Mutual Information-based Losses via Regularization"
_auai.org/UAI/2022/Conference — UAI 2022 Poster_

### Official Review · Reviewer_SpaS · 2022-04-11

**Q2(1) Originality/Novelty:** 3
**Q2(2) Significance/Impact:** 3
**Q2(3) Correctness/Technical Quality:** 3
**Q2(6) Clarity Of Writing:** 4
**Q6 Overall Score:** 6
**Q8 Confidence In Your Score:** 2

**Q1 Summary And Contributions:**

The paper investigates the reasons for variation in mutual information estimation in Mutual Information Neural Estimation (MINE) and other MI estimators. It proposes a  new regularization term for eliminating the variance in the mutual information estimation. Experiments show that the regularization term stabilizes training and decreases the variance. The method is evaluated for supervised and contrastive learning and shows an increase in performance.

**Q2 Assessment Of The Paper:**

More detailed information regarding each of these aspects is given below:

**Q2(4) Quality Of Experiments (Optional):**

3: Good: The experimental evaluation is adequate, and the results convincingly support the main claims.

**Q2(5) Reproducibility:**

2: Fair: Key resources (e.g., proofs, code, data) are unavailable but key details (e.g., proof sketches, experimental setup) are sufficiently well-described for an expert to confidently reproduce the main results.

**Q3 Main Strengths:**

1. The method is well-motivated, and the experiments are well designed to understand the reason for drifting and exploding in the MI estimation.

2. The regularized is simple to implement and solves both drifting and exploding problems.

3. The method is benchmarked for both supervised and contrastive learning tasks, which clearly shows that the method is effective on a real-world dataset.

4. Extensive experiments prove that the proposed regularization term decreases variance.


**Q4 Main Weakness:**

1. The experiments to investigate drifting and exploding behavior are only done on a one-hot toy dataset. It doesn’t guarantee that the same behavior would be observed with the real-world dataset.

2. Theorem 7 doesn’t seem correct. It is mentioned that I(X,Y) = H(Y). This implies H(Y | X) = 0. It is not clear why this assumption holds true.


**Q5 Detailed Comments To The Authors:**

Suggestions:
Remove the overleaf link from the first page!!

Questions:
The first term gets increased by the first term but occasionally decreased by the second term. Why doesn’t the first term in MINE doesn’t decreases in the unstable case?


**Q7 Justification For Your Score:**

The method is well motivated, and extensive experiments show improvement in the performance over baselines.

**Q9 Complying With Reviewing Instructions:**

1: Yes.

---

### Official Review · Reviewer_T3kc · 2022-04-13

**Q2(1) Originality/Novelty:** 3
**Q2(2) Significance/Impact:** 3
**Q2(3) Correctness/Technical Quality:** 3
**Q2(6) Clarity Of Writing:** 3
**Q6 Overall Score:** 5
**Q8 Confidence In Your Score:** 3

**Q1 Summary And Contributions:**

This paper presents an analysis that the second term in the mutual information estimator leads to instability when batch size is small and proposes to regularize the scale of the second term.

**Q2 Assessment Of The Paper:**

More detailed information regarding each of these aspects is given below:

**Q2(4) Quality Of Experiments (Optional):**

3: Good: The experimental evaluation is adequate, and the results convincingly support the main claims.

**Q2(5) Reproducibility:**

3: Good: Key resources (e.g., proofs, code, data) are available and key details (e.g., proofs, experimental setup) are sufficiently well-described for competent researchers to confidently reproduce the main results.

**Q3 Main Strengths:**

1. The paper is well written and clearly explains the existing MI estimators.

2. The analysis is interesting and shows that the second term of the MI estimator is causing the problem.

3. Theoretical analyses are provided to guarantee the correctness of the added regularization.

4. The improvements on CIFAR100 look good and demonstrate the necessity to solve this instability issue.

**Q4 Main Weakness:**

1. Could you further clarify the difference between this paper and the SMILE paper?  According to my understanding, SMILE provides theoretical analysis about the second term while this paper also provides some experiments to show the second term is causing some problems.  SMILE clips the output while the proposed method softly constrains the outputs.

2. Theorem 5 explains that the variance of the second term can be large due to different optimal T. Could you please explain why the variance is still large for the NWJ term? The optimal solution is unique for it.

3. The proposed method constrains the square of the second term rather than clipping the output. What would happen if the optimal T needs to have a large scale? Do you have some possible solutions for how to select the C^*?

I can raise the score if the above questions are addressed, in particular, the first one.

**Q5 Detailed Comments To The Authors:**

as in above

**Q7 Justification For Your Score:**

This paper is interesting but some further clarification of the difference between the SMILE and this paper are needed.

**Q9 Complying With Reviewing Instructions:**

1: Yes.

---

### Official Review · Reviewer_otAn · 2022-04-20

**Q2(1) Originality/Novelty:** 2
**Q2(2) Significance/Impact:** 2
**Q2(3) Correctness/Technical Quality:** 2
**Q2(6) Clarity Of Writing:** 2
**Q6 Overall Score:** 5
**Q8 Confidence In Your Score:** 3

**Q1 Summary And Contributions:**

The paper identifies two causes of the instability of the MI estimators, drifting and output exploding. The paper proposes two regularized estimators ReNWJ and ReDV to combat the instability. The effectiveness of the proposed methods is tested in supervised and contrastive learning. Empirically study on CIFAR-10/-100 support the claim.

**Q2 Assessment Of The Paper:**

More detailed information regarding each of these aspects is given below:

**Q2(4) Quality Of Experiments (Optional):**

2: Fair: The experimental evaluation is weak: important baselines are missing, or the results do not adequately support the main claims.

**Q2(5) Reproducibility:**

2: Fair: Key resources (e.g., proofs, code, data) are unavailable but key details (e.g., proof sketches, experimental setup) are sufficiently well-described for an expert to confidently reproduce the main results.

**Q3 Main Strengths:**

- methodology: the method proposed is well motivated. It looks simple but effective.

- theoretical analysis: the paper provides a detailed analysis of the proposed method.

**Q4 Main Weakness:**

- experiment scale is small: The paper's experiments are mainly on small datasets CIFAR-10/-100. Typically, people use larger datasets (i.e. ImageNet, STL-10) when evaluating contrastive learning algorithms.

- more details of method design: The author directly proposes the method of regularizing the second term of the dual form without explaining why make this choice


**Q5 Detailed Comments To The Authors:**

I will focus more on the weakness.

- method: I am curious why the author chooses to regularize the second term of the dual form. How about we regularized the first term? Also, if the L_2 norm of the neural network's parameters is bounded, then the neural network's output is implicitly bounded. So could we just use weight decay as a tool to stabilize the MI estimation? Of cause one might need to explore what is the best way to perform weight decay, e.g., should apply it to lower layers or higher layers?

**Q7 Justification For Your Score:**

I find the paper shows some novel ideas. The method is well motivated. The experiment supports the claims although it could be better if the author verify their findings on larger datasets.

**Q9 Complying With Reviewing Instructions:**

1: Yes.

---

### Decision · Program_Chairs · 2022-05-15

**Decision:**

Accept (Poster)

**Comment:**

Meta Review: This paper studies the instability of neural network-driven mutual information estimators. It identifies two reasons for instability including the non-convergence of neural networks and the saturating neural network outputs. Then a practical regularization is proposed to mitigate the instability issues. Both theoretical and experimental studies are conducted to demonstrate the effectiveness of the regularizer.

All the reviewers agree that the proposed method is novel, simple, and effective. The theoretical analysis is insightful and the empirical results are good. Nevertheless, there are some concerns on the small scale of datasets. Given that the novelty and technical contribution of this paper outweighs the concerns, I recommend acceptance of this paper. But I highly suggest the authors add results of larger scale experiments (for example Imagenet) in the final version.